# Characteristics of Hearing Loss in Patients with Systemic Lupus Erythematosus

**DOI:** 10.3390/jcm11247527

**Published:** 2022-12-19

**Authors:** Huixian Chen, Fan Wang, Ye Yang, Bingzhu Hua, Hong Wang, Jie Chen, Xuebing Feng

**Affiliations:** 1Department of Rheumatology and Immunology, Nanjing Drum Tower Hospital, Medical School of Southeast University, Nanjing 210008, China; 2Department of Otolaryngology, Nanjing Drum Tower Hospital, Medical School of Southeast University, Nanjing 210008, China

**Keywords:** systemic lupus erythematosus, hearing loss, secondary Sjögren’s syndrome, morbidity, risk factors

## Abstract

Objective: To investigate the clinical characteristics of hearing loss (HL) in patients with systemic lupus erythematosus (SLE) and its related factors. Methods: Ninety-one hospitalized SLE patients and thirty healthy controls were enrolled. All subjects completed pure tone audiometry (PTA), extended high frequency audiometry (EHFA) and distortion product otoacoustic emission (DPOAE) to assess hearing function. SLE patients were divided into two groups according to the presence or absence of HL, and the risk factors of HL were determined by multivariate logistic regression. Results: The incidence of HL was 27.47% in SLE patients, significantly higher than in the control group (3.3%) and most cases were mild-to-moderate, bilateral and predominantly sensorineural. Compared with the control group, the hearing thresholds of SLE patients increased significantly in the middle and high frequencies starting from 2000 Hz. Even though the PTA test results were normal, the EHFA test results showed significant differences in hearing impairment between SLE patients and normal controls. For patients with abnormal PTA results, the signal-to-noise ratio (SNR) in DPOAE was markedly reduced, and the pass rate was also decreased. The Systemic Lupus International Collaborating Clinics Damage Index (SDI, OR 9.13) and secondary Sjögren’s syndrome (sSS, OR 8.20) were identified as independent associated factors for HL, and there was no difference in PTA and EHFA at all frequencies between hydroxychloroquine users and non-users. Conclusions: HL is not rare in SLE patients, and EHFA can help identify early hearing impairment. Having a high SDI score and secondary Sjögren’s syndrome may predict the presence of HL in SLE.

## 1. Introduction

Since McCabe introduced the concept of “autoimmune sensorineural hearing loss” in 1979 [1], both rheumatologists and otolaryngologists have paid increasing attention to the relationship between autoimmune diseases and hearing loss (HL). Although there are conflicting data regarding the mechanisms of HL in the literature, its autoimmune background can be strengthened by the fact that acute sensorineural HL can be adequately treated using steroid treatment in different ways, such as intravenous or intratympanic administration [2,3]. HL is a pathological change of decline in auditory sensitivity of the ear, defined as the average hearing threshold ≥ 20 dB at 500, 1000, 2000 and 4000 Hz frequencies by pure tone audiometry (PTA) [4]. According to the lesion site, HL can be divided into three different types, namely conductive hearing loss (CHL), sensorineural hearing loss (SNHL) and mixed hearing loss (MHL) [4]. Among these, severe hearing loss may influence communication and even result in depression or other serious mental illness, so as to affect the patients’ quality of life.

Systemic lupus erythematosus (SLE) is a prototypic autoimmune disease with multi-system involvement and diverse clinical manifestations that has previously been reported to cause hearing impairment, of which SNHL is most common [5]. Some patients may also present vestibular and auditory symptoms including tinnitus and vertigo [5,6]. It is speculated that HL may be related to the deposition of immune complexes in the stria vascularis and endolymphatic sacs and cell-mediated cytotoxic damage to cochlear and vestibular hair cells [5]. However, most of the current studies on the characteristics of HL in SLE have small sample sizes and inconsistent results, and the influencing factors and associated mechanisms remain unclear. For example, one study found that the hearing level of patients with lupus nephritis was lower than that of patients without lupus nephritis, and the use of hydroxychloroquine (HCQ) might aggravate hearing impairment in SLE [7], while another study showed that high prevalence of HL in SLE was not affected by antimalarial drug use [8].

The aim of this study is to confirm the relationship between SLE and HL and search for the related risk factors and possible mechanisms. We applied extended high frequency audiometry (EHFA) and distortion product otoacoustic emission (DPOAE) examinations in addition to conventional pure tone audiometry (PTA). EHFA is a method of hearing threshold detection at a higher frequency than conventional methods, which can detect hearing impairment early [9], and DPOAE may reflect the function of the outer hair cells of the cochlea [10]. In this study, the effects of antimalarial drug use in hearing function were also evaluated. 

## 2. Materials and Methods

### 2.1. Study Design

A total of 91 SLE inpatients between May 2019 and February 2020 in the Department of Rheumatology and Immunology, Nanjing Drum Tower Hospital were enrolled, and 30 age and gender matched healthy volunteers were enrolled as normal controls. Subjects in both the SLE and control groups were between 18 and 65 years of age. All the patients fulfilled the updated American College of Rheumatology criteria for classification of SLE [11] or the 2019 European League Against Rheumatism/American College of Rheumatology (EULAR/ACR) systemic lupus erythematosus (SLE) classification criteria [12].

Exclusion criteria for SLE patients were: (1) history of traumatic brain injury; (2) exposure to a noise above 85 dB for at least 8 h every day for more than three consecutive months; (3) long-term use of ototoxic drugs, such as streptomycin, gentamicin, vincristine and cisplatin; (4) history of central nervous system involvement not related to SLE, such as encephalitis or a brain tumor; (5) history of diabetes mellitus; (6) persistent poorly controlled hypertension for more than one year; (7) ear disease or family history, such as Ménière’s disease or atresia aural; and (8) concomitant other primary autoimmune diseases except secondary Sjogren’s syndrome (sSS).

### 2.2. Hearing Examination

Audiometry: Audiometric examination is a subjective examination to evaluate the ability to perceive sounds, including PTA and EHFA. The pure tone (air and bone) hearing frequency refers to conventional frequency including 250, 500, 1000, 2000, 4000 and 8000 Hz, and the extended high frequency includes 9000, 10,000, 11,200, 12,500, 14,000 and 16,000 Hz. All tests were performed in a closed audiometry room by the same doctor in the hospital and audiometers (Madsen Astera, Otometrics, Copenhagen, Denmark) were calibrated according to national standards. We adopted the masking rules from the American Speech–Hearing Association while testing our patients [13]. The hearing threshold of each frequency was recorded and for subjects whose hearing threshold could not be measured by the maximum output at a certain frequency, the maximum output plus 5 dB was used as their hearing threshold during statistics.

According to the latest World Hearing Report published by WHO in March 2021 [4], the degrees of HL were classified as the hearing threshold in the better hearing ear: mild (the average hearing threshold ≥ 20 dB and <35 dB), moderate (≥35 dB and <50 dB), moderate severe (≥50 dB and <65 dB), severe (≥65 dB and <80 dB), profound (≥80 dB and <95 dB) and complete (≥95 dB). Unilateral HL was defined as <20 dB in the better ear, while 35 dB or greater in the worse ear. The average hearing threshold of both ears differing by 15 dB was considered asymmetric HL. So far, there were no clear diagnostic criteria for extended high frequency hearing loss; thus, hearing thresholds at each frequency were judged by comparison with healthy controls.

DPOAE: The measurements of DPOAE (Interacoustics Titan IMP440, Middelfart, Denmark) were performed in an acoustically and electrically shielded room. The DPOAE response to pairs of primary tones was measured (f1 and f2; f2/f1 = 1.2), and the DPOAE amplitude was determined as the level of the two f1–f2 components and analyzed for f2 at 684, 988, 1481, 2222, 2963, 4444, 5714 and 8000 Hz. The responses of DPOAE were considered present at each frequency if the response amplitude was ≥6 dB above noise level (signal-to-noise ratio, SNR).

### 2.3. Data Collection

Patients’ clinical data including gender, age, disease duration, disease activity and damage, organ involvements, laboratory findings and treatments were documented. Disease activity was calculated according to the SLE Disease Activity Index (SLEDAI) score [14] and organ damage was determined by the Systemic Lupus International Collaborating Clinics (SLICC)/American College of Rheumatology (ACR) Damage Index (SDI) [15]. For the definition of specific organ involvement and the normal value of laboratory tests, we referred to our previously published paper [16].

### 2.4. Statistics

Data were analyzed using SPSS Statistics 24.0 software (IBM Corp, Armonk, NY, USA). Quantitative values were expressed as mean ± standard deviation or median (interquartile) and analyzed by the *t*-test or the Mann–Whitney U Test, depending on whether the data were normally distributed or not (K–S test). The Kruskal–Wallis test was used to detect the difference among three groups. Enumeration data were presented as percentages and compared by the chi-squared test. Variables significantly related to HL in univariate analyses (*p* ≤ 0.05) were entered into the multivariate logistic regression model to determine their independency, with missing data treated as normal values, and results were reported as odds ratio (OR) with 95% confidence intervals (CI). *p* < 0.05 was considered statistically significant.

## 3. Results

### 3.1. Demographics of the Patients

Of the patients enrolled, 82 (90.1%) were female and the average age was 37.3 ± 13.2 years, with a disease duration of 6.2 ± 6.6 years. The SLEDAI score at the time of hearing examination was 6.0 ± 3.9, while 57 (62.6%) patients had an SDI ≥1 and 28 (30.8%) patients had an SDI ≥ 2. Most patients had renal, cardiopulmonary and hematological involvements, and 16 (17.6%) of them were complicated with sSS (Table 1).

### 3.2. Hearing Thresholds at Different Frequencies in SLE Patients

Compared with the control group, the hearing thresholds of the left ears in the SLE group were statistically significantly different (*p* < 0.05) in each frequency, and the hearing thresholds of the right ears were also statistically significantly different (*p* < 0.05) except in the frequencies of 250, 500 and 2000 Hz (Figure 1). HL was detected in 25 (27.47%) SLE patients, including twenty sensorineural, four mixed and one conductive HL, higher than that of control group in which only one person (3.3%) suffered from HL (*p* < 0.01). Among the patients with HL, two (8%) were unilateral, five (20%) were bilateral asymmetric and eighteen (72%) were bilateral symmetric. When classified by severity, six patients (24%) had moderate HL and seventeen patients (68%) had mild HL. Only four patients presented subjective symptoms, including tinnitus (*n* = 1), vertigo (*n* = 1) and decreased auditory acuity (*n* = 2), and all of them had HL by PTA, of which three had moderate HL.

### 3.3. EHFA Detection for Patients with Normal PTA

Similar results to PTA were seen in the extended high-frequency detection (9000–16,000 Hz) (Figure 1). To further clarify the clinical significance of EHFA, we compared the extended high-frequency hearing thresholds of patients without HL in the SLE group with those of age- and gender-matched healthy controls. All of the patients aged between 20 and 40 years, with a total of 38 ears in the SLE group and 16 ears in the control group, were included. There were statistical differences between the two groups at 11,200, 14,000 and 16,000 Hz (*p* < 0.05) (Figure 2), suggesting that SLE patients might have subclinical hearing loss when the hearing thresholds at the conventional frequencies were still normal, and EHFA is more sensitive for early detection.

### 3.4. Association of DPAOE Results with PTA

In order to search for the etiology of HL in SLE patients, we performed the DPAOE test and correlated the results with the PTA. The data were divided into two groups based on whether a single ear PTA test was normal (122 ears) or not (59 ears). As shown in Figure 3, both SNR values and percentages of valid DPOAE at each frequency between the two groups was statistically different, and those having abnormal results of PTA had lower SNR and percentages of valid DPOAE, suggesting a link between outer hair cell impairment and HL in SLE patients.

### 3.5. Factors Related to HL in SLE Patients

Next, patients’ clinical and laboratory variables were categorized, and logistic regression was used to search for factors associated with HL. According to univariate analysis, patients older than 45 years, with a disease duration > 8 years, an SDI score ≥ 2, complicated with sSS and with a decreased glomerular filtration rate (GFR) were associated with the presence of HL (*p* ≤ 0.05). These variables were then included in multivariate logistic regression analysis, and the results showed that the obtained logistic model had statistical significance (*p <* 0.001), and only two variables were identified to be independently linked to HL, which were SDI score ≥ 2 (OR 9.13) and complicated with sSS (OR 8.20) (Table 2).

### 3.6. Characteristics of HL with Different Associated Factors

Compared with those with an SDI score < 2, patients with an SDI score ≥ 2 had significantly higher hearing thresholds at 1000, 2000, 4000, 8000 Hz and all extended high frequencies in left ears, and higher hearing thresholds at 2000, 4000 Hz and all extended high frequencies in right ears (*p* < 0.05) (Figure 4A,B). Meanwhile, patients with combined sSS had significantly higher hearing thresholds at 2000 and 9000 Hz in their left ears *(p <* 0.05) and higher hearing thresholds at 2000 Hz in their right ears (*p* < 0.05) (Figure 4C,D). Thus, it is suggested that the HL of patients with serious cumulative disease damage was more widely involved, while those with sSS was mainly concentrated in the middle frequencies.

### 3.7. Influence of Hydroxychloroquine on Hearing

In terms of therapeutic drugs, there is currently concerns about the effects of antimalarial drugs on HL [16]. In this study, 64 patients had been treated with hydroxychloroquine (HCQ) for an average of nearly 4 years. Among them, 18 (28.13%) developed HL, which was not statistically different to those not using HCQ (7/27, 25.93%) by the chi-squared test (*p* > 0.05). The average daily dose of HCQ in SLE patients with HL was 216.00 ± 159.19 mg/d, similar to that in SLE patients without HL (247.69 ± 173.74 mg/d). The comparison of hearing thresholds at each frequency in SLE patients using or not using HCQ is shown in Table 3. There was no significant difference in the hearing threshold at each frequency between HCQ users and nonusers, even at the extended high-frequencies. We also checked the relationship between average daily dose of HCQ and HL, and patients who continuously used HCQ for over 1 year were selected and divided into two groups, including daily dose < 400 mg (*n* = 19, 38 ears) and ≥ 400 mg (*n* = 11, 22 ears), and compared with those who did not use HCQ (*n* = 27, 54 ears). Consistently, HL in patients taking a full dose of HCQ (0.4 g/day) was no worse than those not taking HCQ. 

## 4. Discussion

In this study, we showed that the incidence of HL in patients with SLE was 27.47%, among which the majority were sensorineural (80%), consistent with results previously reported [5]. HL in SLE mainly presented in a bilateral and symmetrical distribution and predominantly affected the high-frequency ranges without subjective symptoms, while EHFA could identify auditory damage earlier than PTA. Based on the DPAOE test, HL in SLE was associated with outer hair cell damage. Interestingly, our data demonstrated for the first time that SLE patients with a high SDI score or secondary SS were more likely to have a hearing impairment, and the effects on hearing frequencies were varied.

The incidence of HL in lupus varies widely in the literature, ranging from 6% to 35% [5]. It can progress slowly or suddenly, mainly involving middle and high frequencies, similar to typical age-related deafness [5,17,18,19]. Conventional PTA has great limitations in the detection of early hearing damage, while EHFA may detect the early lesions of the cochlea [9]. In this study, we assessed hearing of very high frequencies of 9000–16,000 Hz in SLE patients with normal hearing thresholds in conventional frequencies compared with healthy controls and observed a statistical difference at 11,200, 14,000 and 16,000 Hz between the two groups. The average hearing threshold at each extended high frequency in the SLE patients was higher than that of the controls, indicating that some lupus patients already have impaired cochlear function before they have reached HL. Consequently, it would be necessary to popularize extended high frequency audiometry to identify early hearing impairment before the patients develop irreversible auditory deficits.

DPOAE refers to the audio energy generated by stimulating the cochlea to pure tone with a certain intensity ratio relationship, which can be recorded in the external auditory canal. This test is able to recognize early cochlear damage as it can reflect the healthy inner environment of the cochlea and integrity of outer hair cells [20]. Our data showed that SNR and passing rates at all frequencies were significantly lower in abnormal PTA, supporting that the outer hair cell injury of the cochlea in SLE patients may contribute to HL. Consistent with our results, current studies have revealed that SLE patients had moderate to severe cochlear hair cell damage, mainly outer hair cell damage, and the deposits of immune complexes in the stria vascularis could cause arterial microvasculature ischemia of the basal turn of the cochlea, eventually leading to hair cell damage [5,21,22,23].

Our study identified that the risk of HL in SLE patients with an SDI ≥ 2 is more than nine times higher than in those with an SDI < 2, suggesting the degree of hearing impairment may reflect a patient’s cumulative organ damage. Meanwhile, patients with SLE and secondary SS had a 7.2-fold increased risk of HL. SS is an immune disorder primarily affecting the exocrine glands and previous studies have found that primary Sjörgen’s syndrome also causes hearing loss by causing dryness in the ear canal epidermis, middle and inner ear fluids [24]. The change in the volume of the inner ear fluids can disrupt the molecular structure of the fluids, which may injure the spiral ganglion cell and cochlear hair cell [24]. Similarly, sSS may also cause hearing impairment by the same mechanism and may accelerate the process of hearing loss in SLE. In addition, in a small sample report [7], hydroxychloroquine was considered to be associated with hearing loss in SLE patients. Recently, slight hearing loss has been reported in patients with COVID-19 with or without HCQ [25]. Polanski et al. also analyzed the role of antimalarials on hearing function in SLE patients [8]; in contrast, they found a statistically worse performance in antimalarial drug nonusers versus users at 8000 Hz but not at other frequencies. Thus, they concluded that a high prevalence of HL in SLE is not affected by antimalarial drug use. Here, we first examined the effect of hydroxychloroquine on hearing at extended high frequencies, and showed that the antimalarial drug was not related to hearing impairment. The reason why HCQ does not cause HL in this paper may be related to two reasons; one is the therapeutic effect of HCQ on diseases, and the other is the low drug concentration in Chinese patients. Recently, we have shown antimalarial drugs were associated with lower risk of SLE mortality, and the protective effects for survival might be augmented by adherence and full dosage of these drugs [26]. Meanwhile, our preliminary study shows that even if HCQ is taken in a full dosage (0.4/day), its blood concentration in Chinese patients is still significantly lower than that in patients from Western countries [27].

Ferrari et al. [28] found asymptomatic SNHL in 16% of SLE and associated it with the levels of low-density lipoprotein, suggesting atherosclerosis could be a mechanism for HL in SLE. However, this phenomenon has not been confirmed in this study. As we have initially excluded SLE patients with diabetes and hypertension that had been poorly controlled for more than one year, this may lead to the elimination of some risk factors of atherosclerosis, thus weakening its association with HL.

There is still much potential for improvement of this study, for example, by investigating the reversibility of hearing loss in patients with SLE after treatment and by performing comprehensive hearing tests, including tympanometry, auditory brainstem response (ABR) and brain MRI, to rule out other influences, including new otitis media, on conductive HL and central complications. Moreover, the sample sizes used to compare the efficacy between EHFA and PTA tests, as well as the impact of HCQ on HL, were small, which may cause bias. Nevertheless, our data indicate that sensorineural HL is prevalent among SLE patients and EHFA should be recommended to identify early hearing impairment. High SDI and secondary SS are independently related to HL in SLE, while hydroxychloroquine has no influence on hearing function.

## Figures and Tables

**Figure 1 jcm-11-07527-f001:**
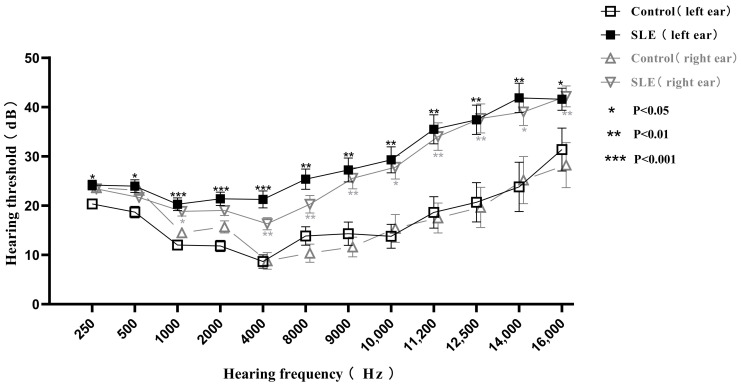
Comparison of pure tone audiometry and extended high frequency audiometry thresholds from 250 to 16,000 Hz frequencies between 91 SLE patients and 30 healthy controls. Data are presented as mean ± SEM and the mean values at each hearing frequency are linked by the trend line. * *p* < 0.05, ** *p* < 0.01, *** *p* < 0.001 by *t*-test.

**Figure 2 jcm-11-07527-f002:**
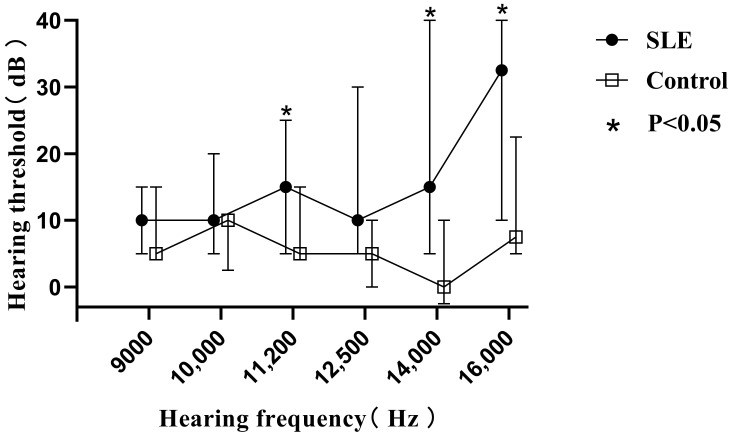
Comparison of extended high frequency audiometry thresholds at 9000 to 16,000 Hz between SLE patients (38 ears) and controls (16 ears) with normal pure tone audiometry thresholds. Data are presented as median and interquartile range, and the median values at each hearing frequency are linked by the trend line. * *p* < 0.05 by Mann–Whitney U Test.

**Figure 3 jcm-11-07527-f003:**
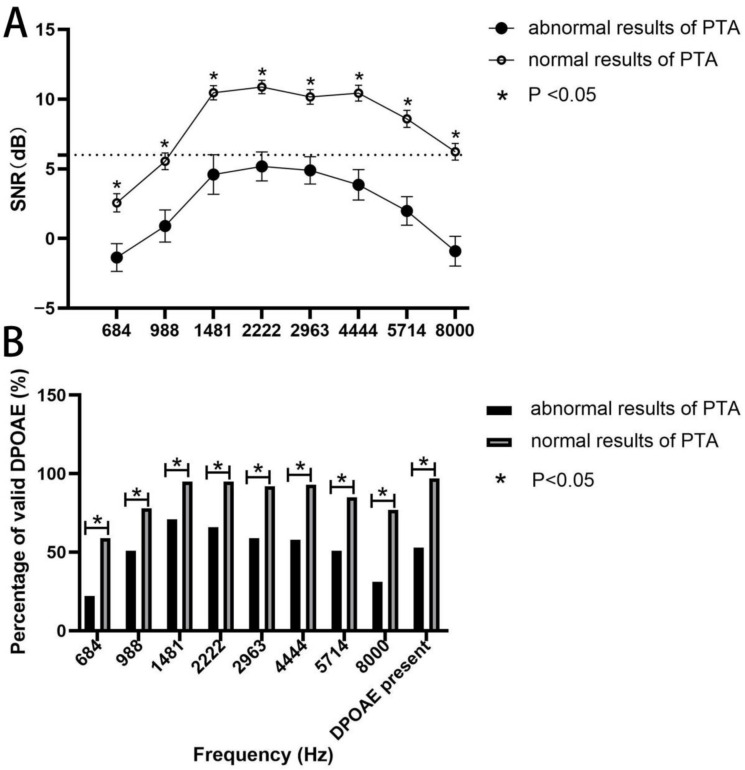
Association of DPOAE results with PTA. (**A**) SNR values in the ears of SLE patients with and without normal PTA at different frequencies. The dashed line indicates normal SNR value (6 dB). Data are presented as mean ± SEM and the mean values at each F2 frequency are linked by the trend line. *p* < 0.05 by *t*-test. (**B**) Percentages of valid DPOAE in the ears of SLE patients with and without normal PTA at different frequencies. *p* < 0.05 by chi-squared test. DPOAE: distortion product otoacoustic emission; PTA: pure tone audiometry; SNR: signal-to-noise ratio.

**Figure 4 jcm-11-07527-f004:**
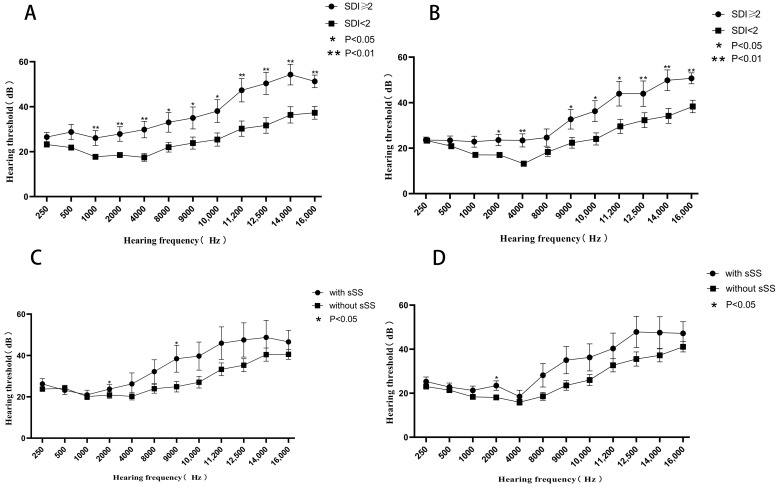
Hearing thresholds at different frequencies in SLE patients. (**A**,**B**) Comparison of hearing thresholds in SLE patients with SDI ≥ 2 (*n* = 28) and <2 (*n* = 63) in the left or right ear. (**C**,**D**) Comparison of hearing thresholds in SLE patients with (*n* = 16) and without sSS (*n* = 75) in the left or right ear. Data are presented as mean ± SEM and the mean values at each hearing frequency are linked by the trend line. * *p* < 0.05, ** *p* < 0.01 by *t*-test. SDI: Systemic Lupus International Collaborating Clinics/ACR damage index, sSS: secondary Sjögren’s syndrome.

**Table 1 jcm-11-07527-t001:** Demographics of SLE patients and controls.

Features	SLE Patients	Healthy Controls
Age (years)	37.26 ± 13.17	35.70 ± 12.22
Gender (female)	82 (90.1%)	24 (80.0%)
Disease duration (years)	6.23 ± 6.64	
SLEDAI score	6.0 ± 3.9	
SDI score ≥ 1	57 (62.6%)	
Complicated with Sjögren’s syndrome	16 (17.6%)	
Organ involvement		
Mucocutaneous	44(48.4%)	
Musculoskeletal	34(37.4%)	
Renal	63(69.2%)	
Cardiopulmonary	57(62.6%)	
Gastrointestinal	26(28.6%)	
Hematological	47(51.6%)	
Neuropsychiatric	12(13.2%)	
Vertigo	1(1.1%)	
Tinnitus	1(1.1%)	
Decreased auditory acuity	2(2.2%)	

Data are shown as mean ± SD or number (percentages).

**Table 2 jcm-11-07527-t002:** Factors related to hearing loss in SLE patients (by logistic regression).

Factors		Univariate			Multivariate		
		OR	95%CI	*p*	OR	95%CI	*p*
Age > 45 years		6.64	2.42–18.26	<0.001	3.132	0.93–10.56	>0.05
Gender (female/male)		0.73	0.17–3.19	>0.05			
Disease duration >8 years		2.89	1.11–7.50	0.026	2.362	0.65–8.61	>0.05
SLEDAI > 10		2.46	0.80–7.56	>0.05			
SDI ≥ 2		6.12	2.24–16.69	<0.001	9.13	2.34–35.65	0.001
Secondary Sjögren’s syndrome		4.74	1.53–14.70	0.011	8.20	1.713–39.30	0.008
Hypertension ^#^		2.29	0.83–6.34	>0.05			
Organ involvements	Mucocutaneous	0.98	0.39–2.46	>0.05			
	Musculoskeletal	1.17	0.45–3.00	>0.05			
	Renal	1.58	0.55–4.53	>0.05			
	Cardiopulmonary	1.78	0.65–4.85	>0.05			
	Gastrointestinal	1.62	0.61–4.34	>0.05			
	Hematological	1.02	0.41–2.56	>0.05			
	Neuropsychiatric	2.11	0.60–7.39	>0.05			
Lab tests	White blood cell < 4 × 10^9^/L	1.16	0.43–3.15	>0.05			
	Platelet < 100 × 10^9^/L	1.25	0.39–4.05	>0.05			
	Creatinine > 133 umol/L	2.07	0.59–7.27	>0.05			
	GFR < 90 mL/min/1.73 m^2^	2.65	0.98–7.14	0.05	0.98	0.24–3.99	>0.05
	C3 < 0.8 g/L	0.45	0.18–1.14	>0.05			
	C4 < 0.2 g/L	0.50	0.18–1.43	>0.05			
	Proteinuria > 0.5 g/24 h	1.42	0.52–3.91	>0.05			
	ANA (+)	0.89	0.16–4.93	>0.05			
	Anti-Sm antibody (+)	1.29	0.40–4.24	>0.05			
	Anti-dsDNA antibody (+)	0.89	0.26–3.01	>0.05			
	Anti-cardiolipin antibody (+)	3.00	0.54–16.60	>0.05			
	Anti-glycoprotein antibody (+)	0.78	0.08–7.89	>0.05			
	Cholesterol < 5.72 mmol/L	0.86	0.25–2.96	>0.05			
	LDL < 3.1 mmol/L	1.28	0.44–3.70	>0.05			
	HDL > 0.94 mmol/L	2.28	0.84–6.18	>0.05			
Treatments	Glucocorticoid	0.37	0.02–6.14	>0.05			
	Cyclophosphamide	0.65	0.19–2.18	>0.05			
	Hydroxychloroquine	0.83	0.40–3.01	>0.05			
	Tacrolimus	1.44	0.51–4.14	>0.05			

OR: Odd Ratio; CI: Confidence Intervals; SLEDAI: Systemic Lupus Erythematosus Disease Activity Index; SDI: Systemic Lupus International Collaborating Clinics/ACR damage index; GFR: Glomerular Filtration Rate; C3: Complement C3; C4: Complement C4; LDL: Low-density lipoprotein; HDL: High-density lipoprotein. ^#^ Those poorly controlled for more than one year has been ruled out.

**Table 3 jcm-11-07527-t003:** Hearing threshold at each frequency in SLE patients using and not using HCQ.

Hearing Frequency (Hz)	With HCQ, 128 Ears	Without HCQ, 54 Ears	*p*
250	25 (20, 25)	20 (20, 25)	0.766
500	20 (20, 25)	20 (15, 25)	0.353
1000	15 (10, 25)	15 (10, 20)	0.537
2000	20 (15, 25)	15 (10, 25)	0.202
4000	15 (10, 20)	15 (10, 25)	0.703
8000	15 (10, 30)	15 (10, 30)	0.649
9000	17.5 (10, 40)	15 (10, 35)	0.930
10,000	20 (10, 45)	20 (10, 30)	0.635
11,200	30 (15, 55)	20 (10, 45)	0.157
12,500	40 (15, 65)	25 (10, 50)	0.084
14,000	45 (15, 70)	30 (10, 60)	0.079
16,000	52.5 (30, 60)	47.5 (25, 60)	0.190

## Data Availability

Data are available on request from the authors.

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
