# Peer review of "Characteristics of Hearing Loss in Patients with Systemic Lupus Erythematosus"

_jcm, 2022, doi:10.3390/jcm11247527_

Round 1

Reviewer 1 Report

First of all, I would like to thank you for reviewing this manuscript. The topic of this manuscript is interesting, and the presented results add to knowledge; however, there are necessary changes before it can be published. See my specific comments below.

Abstract, line 23. Presenting this result, it is also essential to state that the appearance of hearing loss was significantly higher than in the control group.

Abstract, line 26. .. the EHFA test showed significant differences between lupus patients and controls. This result refers to what? Please specify.

Keywords. From my point of view, hearing loss must be included here, as this is the main topic of the present article. Furthermore, it should also be mentioned that the secondary Sjögren syndrome was investigated in the present study.

Introduction, line 40. Regarding the autoimmune background of hearing loss, conflicting data in the literature can be observed. However, the autoimmune background can be strengthened by the fact that acute sensorineural hearing loss can be adequately treated using steroid treatment in different ways (e.g., intravenous or intratympanic administration). From my point of view, this fact should be mentioned in the Introduction, including the following reference articles:

[Rozbicki P, Usowski J, Siewiera J, Jurkiewicz D. The Influence of Steroid Therapy on the Treatment Results in Patients with Sudden Sensorineural Hearing Loss. J Clin Med. 2022 Oct 15;11(20):6085. doi: 10.3390/jcm11206085.]

[Molnár A, Maihoub S, Tamás L, Szirmai Á. Intratympanically administered steroid for progressive sensorineural hearing loss in Ménière's disease. Acta Otolaryngol. 2019 Nov;139(11):982-986. doi: 10.1080/00016489.2019.1658898.]

Line 46. Hearing loss can also significantly influence the patients’ quality of life.

Line 82. Please correct the spelling of Ménière's disease.

Line 86. Please include the name and type of the audiometer, along with the manufacturer’s name, the manufacturing city and country, etc. Additionally, I also miss some technical information regarding the audiometry examination (e.g., air-and bone conduction measurements, masking, etc.). It is also not stated whether before audiometry, the patients’ external-and middle ear status, tympanometry was performed or not. Using which method were retrocochlear lesions ruled out? Was ABR and/or brain MRI performed?

Line 104. Please include the type of the OAE device.

Line 119. Please include the manufacturer, manufacturing city, and country of the SPSS software.

Line 121. Based on which normality test was the normal distribution analysed?

Table 1. I think it would be interesting to include the ratio of hearing loss and tinnitus as subjective complaints.

Lines 143-145. To what are these statistically significant differences referring? Please explain. I also miss the p-values here if the authors are talking about statistical differences.

Line 151. Regarding vertigo, were clinical data accessible? Any specific diagnosis behind the vertigo symptom? I also recommend including this in Table S1. Regarding tinnitus: this tinnitus case could be categorised into secondary cases of tinnitus with a possible autoimmune background. I think this (i.e., secondary tinnitus can also be a concomitant symptom) should be included in the introduction, including the following reference articles:

[Di Stadio A, Ralli M. Systemic Lupus Erythematosus and hearing disorders: Literature review and meta-analysis of clinical and temporal bone findings. J Int Med Res. 2017 Oct;45(5):1470-1480. doi: 10.1177/0300060516688600.]

[Mavrogeni P, Maihoub S, Tamás L, Molnár A. Tinnitus characteristics and associated variables on Tinnitus Handicap Inventory among a Hungarian population. J Otol. 2022 Jul;17(3):136-139. doi: 10.1016/j.joto.2022.04.003.]

Line 164. In this analysis, 38 and 16 cases were contrasted in the two groups; therefore, there was a relatively limited number of control cases. This is a limitation which must be stated at the end of the manuscript.

Figure 3. Please improve the quality of this Figure because it is not easy to read data in its current form.

Table 1. Please explain the abbreviations (i.e., CI and OR).

Figure 4. Please improve the quality of this Figure because it is not easy to read data in its current form.

Line 215. I do not recommend using the abbreviation in the title.

Line 218. ..not statistically different.. – Based on which statistical test and significance level?

Line 220. If the authors state the daily doses of HCQ, the correlation between the dose of the medicine and hearing loss should be calculated and presented.

Table 2. The number of control cases is relatively low (130 vs 52); it must be stated as a limitation.

Discussion – Is there any previous literature data on the ratio of sensorineural, conductive and mixed hearing loss in SLE? Is there any possible explanation for cases of conductive / mixed hearing loss?

In the end of the manuscript, some limitations are missing (see my previous comments). Furthermore, based on the methods, ABR and brain MRI were not performed (or if they were, it was not stated); therefore, the central complications of SLE could not have been ruled out.

References: I find the reference list limited, but including my suggestions above the list will be acceptable.

Reviewer 2 Report

The citations should use more recent literature, the current cited publications all seem to be quite old.     

Please describe how long HCQ treatment was provided to patients. Further,  literature examples have to be cited whereby the same treatment period of HCG as used in this study has resulted in side effects or chronic conditions.

Discuss why only certain frequencies of patients are affected by HL compared to healthy ones.

Legends of tables should be more informativ and stand for themselves without reading the manuscript.

This is a samll study of 91 paeticipants, please delet in the introduction that it was the aim to do a big study cohort.

Legends of the figures should only describe the experiments, provide all infomration needed to understand the graph by itself, do not write results in the legends. Please, put more information into the legends. Mark significancy more clearly in the figure.

Why is the malaria drug causing HL as published by others, but not in this study? Is SLE protective, did they use longer treatment times...?

Do patients suffering from Sjörgen´s syndrom (SS) develop HL - is that the explanation why it is a factor in SLE with secondary SS? Please, add to the discussion.

Comorbidities of the patients have to be shown in the table. As other autoimmune conditions such as RA or Diabetes are risk factors for HL - have they been excluded?

Please, also add accompagnying symptoms meaning Tinnitus and Vertigo

Please, discuss sudy of others using logistic regression which found low-density lipoprotein levels suggesting atherosclerosis as a mechanism. J Clin Rheumatol 2016. Ferrari et al

Round 2

Reviewer 1 Report

Thank you for the corrections. 

Author Response

Many thanks for careful review.

Reviewer 2 Report

1. The figure legends can still be improved. Please add the type of  statistics used.

2. Please write A, B.... above not below of the graphs

3. Figure 1 does not look like high resolution

4. "We adopted the masking rules from the American 100
Speech-Hearing Association while testing our patients." - please, add the citation.
